# TRANSFERRING JAILBREAK ATTACKS FROM PUBLIC TO PRIVATE LLMS VIA LOCAL PROMPT OPTIMIZATION

## ABSTRACT

Large Language Models (LLMs) demonstrate remarkable capabilities across natural language processing tasks but remain vulnerable to jailbreak attacks, where adversarial inputs are crafted to elicit harmful or undesirable responses. Existing optimization-based attacks often achieve high success rates but are impractical in black-box settings. We focus on a practical scenario in which private LLMs are fine-tuned from public models and accessible only via query APIs, reflecting common real-world deployments. To address this, we propose a two-stage local prompt optimization framework that transfers jailbreak attacks from public to private LLMs. Our method introduces an auxiliary adversarial suffix to align output distributions between the public and target private models, enabling gradient-informed optimization in a purely local setup. Experiments show that our approach achieves high attack success rates on both open-source (Vicuna, LLaMA3) and proprietary models (GPT-4, Claude), and remains effective under diverse fine-tuning regimes, including LoRA-based updates. These results highlight the practical security risks of fine-tuning LLMs and the need for robust defenses, while showing that highly transferable black-box attacks can be executed efficiently without accessing private model parameters.

## 1 INTRODUCTION

The rapid surge in the popularity of Large Language Models (LLMs) has sparked both immense excitement and apprehension. Pretrained LLMs like Meta's Llama Touvron et al.; 2023) and OpenAI's GPT Achiam et al. (2023) are now considered indispensable pillars supporting a wide range of AI applications. In practice, customizing pretrained LLMs for specific use cases through fine-tuning is desirable. For example, HuatuoGPT Zhang et al. (2023) incorporates real-world data from doctors during the supervised fine-tuning phase to develop a large language model tailored for medical consultation. Voyager Wang et al. (2023), an LLM-powered embodied lifelong learning agent in Minecraft, autonomously explores the world, acquires diverse skills, and makes novel discoveries without human intervention.

Given their remarkable proficiency across a wide variety of natural language tasks, LLMs hold the promise of significantly boosting society's productivity by automating tedious tasks and readily providing information. Therefore, it's essential to emphasize the security issues associated with LLMs. One severe threat to LLMs is jailbreak, which stems from the extensive training text corpora containing potentially harmful information. Jailbreak Wei et al. (2024) aims to circumvent security measures surrounding an LLM and may even compromise their alignment safeguards Carlini et al. (2024).

The most effective approach to generating jailbreak attacks involves gradient-based optimization to acquire the adversarial input. For instance, GBDA Guo et al. (2021) utilizes the Gumbel-Softmax approximation trick to ensure differentiable adversarial loss optimization. It employs metrics such as BERTScore and perplexity to maintain perceptibility and fluency during optimization. However, this optimization process requires full access to the model parameters and architecture, necessitating the target model to be in the white-box setting.

Figure 1: Attackers can generate adversarial attacks using optimization-based methods on public LLMs. For private LLMs fine-tuned from these public models with private data, locally fine-tuning the generated attacks can also successfully compromise the private LLM, even if the attackers only have query access to the model. This scheme highlights severe security vulnerabilities in fine-tuned private LLMs.

In our study, we introduce a novel jailbreak framework specifically targeting private LLMs in black-box settings, shown in Fig. 1. Despite the challenges posed by inaccessible fine-tuning data and models, fine-tuned LLMs remain susceptible to severe security breaches. As LLMs evolve, it's imperative for researchers to devise robust jailbreak techniques that rigorously test their resilience, ethical principles, and deployment readiness. Our main claim is that **Fine-tuning LLMs may cause severe security issues**, even when the parameters and fine-tuning data of the fine-tuned LLM remain private and inaccessible. We exemplify this claim through the lens of jailbreak attacks. Specifically, we propose an optimization-based attack generation framework for black-box LLMs by optimizing attacks on the open-source LLM from which the target LLM is fine-tuned. Importantly, if the precise base model is unknown, a general-purpose LLM can be used as a surrogate base, and our method remains effective. Subsequently, we apply local fine-tuning on these generated attacks, enabling them to successfully compromise black-box LLMs with performance comparable to attacks conducted with knowledge of the target LLM's parameters.

In a word, our contributions can be summarized as:

- **Investigating Fine-Tuning Attacks:** We are the first to explore the fine-tuning of attacks in the direction of model fine-tuning. This approach is particularly practical in scenarios where many third parties fine-tune open-source LLMs for their private models, offering a novel perspective compared to current research.

- **Flexible Adversarial Attack Framework:** We introduce several transformations of the proposed adversarial attack framework, highlighting its flexibility and practical significance. These transformations enable adaptability to various attack scenarios, enhancing the framework's utility in real-world applications.

- **Demonstrated Effectiveness:** We have demonstrated the effectiveness of the proposed attack generation framework by achieving a relatively high attack success rate. Notably, our results show that the performance of our approach is comparable to that of white-box LLMs, underscoring its efficacy in generating potent adversarial examples.

## 2 RELATED WORK

Here, we begin by reviewing related works on attacking LLMs, followed by an overview of current research focusing on efficiently fine-tuning LLMs.

### 2.1 ATTACKS AGAINST LANGUAGE MODELS

Here, we investigate inference-time attack methods, categorizing them into two settings: white-box and black-box, to explore their impact on language models.

In the white-box setting Shakeel & Shakeel (2022); Wen et al. (2024); Liu et al. (2022), attackers possess complete access to the model parameters and architecture. For instance, GBDA Guo et al. (2021) leverages the Gumbel-Softmax approximation trick to ensure differentiable adversarial loss optimization, utilizing BERTScore and perplexity metrics to enforce perceptibility and fluency. Additionally, HotFlip Ebrahimi et al. (2018), introduced as an efficient gradient-based optimization method, generates adversarial examples by manipulating the discrete text structure within its one-hot representation.

As a solution for the black-box setting, token manipulation-based attacks Morris et al. (2020); Ribeiro et al. (2018); Jin et al. (2020) entail applying basic token operations, such as replacing tokens with synonyms, to a text input sequence to induce incorrect predictions from the model. HQA-attack Liu et al. (2024) addresses the challenging hard label setting by initially generating an adversarial example and then iteratively replacing original words to minimize the perturbation rate.

## 2.2 LLMs Finetuning

Finetuning large language models has emerged as a highly effective strategy for enhancing their performance. In comparison to full fine-tuning approaches, Parameter Efficient Fine-Tuning (PEFT) Mangrulkar et al. (2022) methods involve freezing most parameters of pre-trained models, yet they can still demonstrate comparable capabilities on downstream tasks. The main efficient fine-tuning methods can be summarized as Adapter-based Tuning Mangrulkar et al. (2022); Poth et al. (2023); Rücklé et al. (2020); Wang et al. (2020); Chen et al. (2022b;a), LoRA Hu et al. (2021); Dettmers et al. (2023); Yu et al. (2024), Prefix Tuning Van Sonsbeek et al. (2023); Li & Liang (2021); Yang & Liu (2021), and Prompt Tuning Jia et al. (2022); Wang et al. (2022); Lester et al. (2021).

## 2.3 LLMs Alignment

LLMs alignment Liu et al. (2023c); Kirk et al. (2024); Ji et al. (2023) refers to the process of ensuring that Large Language Models (LLMs) exhibit behavior that aligns with human values and intentions. This includes characteristics such as being helpful, truthful, ethical, and safe in their interactions and outputs. Alignment ensures that models' behaviors align with human values and intentions. For example, aligned LLMs have safety measures to reject harmful instructions. The most common alignment techniques are Instruction Tuning Zhou et al. (2024); Cahyawijaya et al. (2023) and Reinforcement Learning from Human Feedback (RLHF) Song et al. (2024); Ji et al. (2023). Specifically, Liu et al. Liu et al. (2023a) convert various types of feedback into sequences of sentences to fine-tune the model. Jeremy et al. Scheurer et al. (2023) introduce Imitation Learning from Language Feedback (ILF), a novel approach that leverages more informative language feedback. Stiennon et al. Stiennon et al. (2020) compile a large dataset of human comparisons between summaries, train a model to predict the preferred summary, and use this model to fine-tune a summarization policy through reinforcement learning.

# 3 Proposed Method

## 3.1 Problem Formulation

In this paper, we focus on jailbreaking target language models, which we assume to be private with the following characteristics: 1) The parameters of the target model and the private fine-tuning data are unknown. 2) The target model can be normally inferred and responds to given inputs. 3) For problem formulation, we assume the attacker knows which public LLM the model was fine-tuned from; *however, as our experiments show, even if this information is unavailable, the attacker can use a general-purpose LLM as a surrogate base model and still achieve effective attacks.*

This setting is practical because fine-tuning LLMs on private data results in models that not only generate high-quality text but also possess precise domain knowledge. We define the attackers as follows:

**Attackers' Capability.** We assume that attackers only have the capability to query the target private LLM, denoted as $\mathcal{T}_{\theta_{loc}}$, without access to any information about the model parameters $\theta_{loc}$ or the

corresponding training data $\mathcal{D}$. For problem formulation, we assume that attackers know which public LLM $\mathcal{T}_{\theta_0}$ the private LLM is fine-tuned from; however, as our experiments show, even if this information is unavailable, attackers can use a general-purpose LLM as a surrogate base to effectively optimize attacks. Specifically, the target private network is fine-tuned from the public LLM as $\mathcal{F}(\mathcal{D}) : \theta_{loc} = \arg\min_\theta \mathcal{L}(\mathcal{T}_{\theta_0}, \mathcal{D})$, where $\mathcal{L}(\cdot)$ represents the loss function for fine-tuning.

**Attackers' Objective.** The attackers aim to generate attacks capable of jailbreaking the target private model $\mathcal{T}_{\theta_{loc}}$. Specifically, we focus on prompt-level jailbreaks, where the attackers input the prompt $P$ with the objective of finding a prompt that elicits a response $R = \mathcal{T}_{\theta_{loc}}(P)$ demonstrating undesirable behaviors. More formally, the goal is to solve the following problem:

$$\text{find} \quad P \quad s.t. \quad \text{JUDGE}(P, R) = 1 \tag{1}$$

where JUDGE$(\cdot)$ is a binary-valued function, with 1 denoting that the text pair $(P, R)$ is jailbroken. Considering the difficulties in defining the function JUDGE$(\cdot)$, and following previous work Zou et al. (2023), we define a series of negative responses (e.g., "I'm sorry", "As a language model"). Thus, whether the responses are included in the defined negative responses is used to measure the success of the jailbreak attacks.

In our main focus, we aim to attack the target LLM exclusively, ***without expecting the generated attacks to succeed against the public base LLM***. Although the target model's parameters and fine-tuning data are inaccessible, its close relationship to the known (or surrogate) base model allows us to approximate its behavior through proxy-based optimization. Specifically, the shared initialization or structural similarity between the base and fine-tuned models provides a useful inductive bias that enables transferable gradient-based attacks with appropriate local adaptation. This forms the foundation of our two-stage optimization strategy, described in the next section.

## 3.2 ATTACK GENERATION VIA LOCAL FINE-TUNING

Building upon the previous LLMs attack framework Zou et al. (2023), let the target private LLM be represented as a mapping from input tokens $x_{[1:n]} \subseteq P$ to the distribution of the next token, where the probability of the next token is denoted as $p(x_{[n+1]}|x_{[1:n]}; \theta_{loc})$. The objective of the attack is to generate the $H$-token target sequence $x^*_{[n+1:n+H]}$, leading to subsequent adversarial tokens. For the input tokens $x_{[1:n]}$, we set a fixed-length suffix $s_{[1:l]}$ (with $l < n$) to iteratively update for jailbreaking the target LLM. The rest of the instruction prompt is denoted as $x_{in}$, forming $x_{[1:n]} \leftarrow x_{in} + s_{[1:l]}$. Thus, the optimization problem of the adversarial suffix $s$ can be formulated as follows:

$$s^* = \arg\min_{s_{[1:l]} \in V^{|l|}} \mathcal{L}_a(x_{[1:n]}; \theta_{loc}) = \arg\min_{s_{[1:l]} \in V^{|l|}} -\log p\big(x^*_{[n+1:n+H]}|x_{in} + s_{[1:l]}; \theta_{loc}\big);$$

$$\text{where} \quad p\big(x_{[n+1:n+H]}|x_{in} + s_{[1:l]}; \theta_{loc}\big) = \prod_{i=1}^{H} p\big(x_{[n+i]}|x_{[1:n+i-1]}; \theta_{loc}\big); \tag{2}$$

where $V$ denotes the vocabulary size. The above optimizing objective forces the language model to generate the first few positive tokens, with the intuition that if the language model can be put into a "state" where this completion is the most likely response (e.g., responding with "Sure, here's a script that can ..."), rather than refusing to answer the query, it is likely to continue the completion with the desired objectionable behavior.

In this way, since the instruction prompt $x_{in}$ is the prompt that elicits harmful information, the private LLM tends to refuse to give the positive response. We denote the output sequence as $\tilde{x}_{[n+1:n+H]}$ with the current input. We compute the linearized approximation of replacing the $i$-th token in the adversarial prompt, by evaluating the gradient as:

$$Grad(s_{[i]}) = \nabla_{e_{s_i}} \mathcal{L}_{llm}(s_{[i]}; \theta_{loc}), \quad i \in \{1, 2, ..., l\},$$

$$\mathcal{L}_{llm}(s; \theta_{loc}) = \mathcal{D}ist\big[p(\tilde{x}_{[n+1:n+H]}|x_{in} + s; \theta_{loc}), p(x^*_{[n+1:n+H]})\big], \tag{3}$$

where $e_{s_i} \in \{0, 1\}^V$ is the one-hot vector denoting the current value of the $i$-th token, $p(x^*_{[n+1:n+H]})$ is the target output logit values. The distance function $\mathcal{D}ist$ (we could take the cross entropy loss as an example) measures how closely the model's current output matches the target response $x^*$. By solving the optimization in Eq. 3, we could get the top $K$ substitutes (with the largest negative gradient) for each token in the adversarial suffix $s$.

Given that the attackers only have the capability to query the target model (with the parameters $\theta_{loc}$ remaining unknown), direct optimization-based attack generation with the gradient information on Eq. 3 seems impossible. Recall that the attackers are aware of which public LLM the target model is fine-tuned from, of which we denote the parameters as $\theta_0$, finetuning it with the local data pairs $\mathcal{D} = \{x^{(r)}, u^{(r)}\}_{r=1}^R$ could be denoted as:

$$\theta_{t+1} \leftarrow \theta_t - \eta \nabla_{\theta_t} \mathcal{L}_{llm}(\mathcal{D}; \theta_t),$$

$$\mathcal{L}_{llm}(\mathcal{D}; \theta_t) = \sum_{r=1}^R \mathcal{D}ist\big[p(\tilde{x}^{(r)}|x^{(r)}; \theta_t), p(u^{(r)})\big], \tag{4}$$

where the fine-tuning process primarily focuses on optimizing the weight update $\theta$ to maximize the log-likelihood of the targeted model responses.

We make the following approximation for Eq. 3 in the neighborhood of $x_{in}$:

$$\text{Grad}(s) = \nabla_{e_s} \mathcal{L}_{llm}(s; \theta_{loc}) \approx \nabla_{e_s} \mathcal{L}_{llm}(s + \mathbf{a}; \theta_0),$$

$$\text{s.t.} \quad p(x^{(r)}|x_{in} + \mathbf{a}; \theta_0) \sim p(x^{(r)}|x_{in}; \theta_{loc}), \tag{5}$$

where the gradients $\text{Grad}(s)$ are computed on the public LLM $\theta_0$ using the auxiliary suffix $\mathbf{a}$. Intuitively, the optimization process consists of two steps: first, we align the output distribution of the public model with that of the private target model by finding an auxiliary suffix $\mathbf{a}$; then, we conduct adversarial optimization on $s$ over the aligned public model, effectively simulating white-box access to the target. This two-stage procedure enables surrogate gradient estimation and significantly improves attack effectiveness without requiring access to the target model's parameters.

Suppose for a given input prompt $x_{in}$, we can find a suffix $\mathbf{a}$ that sufficiently aligns the outputs of the target and public LLMs. The approximation in Eq. 5 is justified primarily by the following two reasons:

- The target LLM is fine-tuned from the public LLM using parameter-efficient fine-tuning, which freezes most of the parameters of $\theta_0$. Therefore, we first learn the suffix $a$ to align $p(x^{(r)}|x_{in} + \mathbf{a}; \theta_0)$ with $p(x^{(r)}|x_{in}; \theta_{loc})$, and then calculate the gradients of the $a$-aligned public LLM to approximate those of the target model.

- The gradients $Grad(s)$ are calculated to select a set of possible substitutes for $s$ (details will be provided in a later section), which introduces a certain level of error tolerance.

When attacking LLMs, we assume the instruction prompt $x_{in}$ and the target prompt $x^*_{[n+1:n+H]}$ to be fixed. Finally with the approximation in Eq. 5, Eq. 3 could be iteratively optimized in two steps: 1) we optimize the suffix to make the public and target LLMs alignment with the input $x_{in}$; 2) initialize the adversarial suffix $s$ with $a$ and optimize $s$ for the jailbreak attack. To be specific, when the parameters of the LLM are known, with the greedy coordinate gradient-based search algorithm, the optimal adversarial suffix can be obtained to satisfy Eq. 2. The process could be denoted as:

$$a^{(t)} \leftarrow \arg \min_{a \in Replace\{s^{(t-1)}, Grad(a)\}} \mathcal{D}ist\big[p(\tilde{x}|x_{in} + a; \theta_0), p(\tilde{x}|x_{in}; \theta_{loc})\big],$$

$$s^{(t)} \leftarrow \arg \min_{s \in Replace\{a^{(t)}, Grad(s)\}} \mathcal{D}ist\big[p(\tilde{x}|x_{in} + s; \theta_{loc}), p(x^*)\big], \tag{6}$$

$$\text{where} \quad s^0 \leftarrow Random\_Initialize(V^l), \quad \text{and} \quad 1 \le t \le T,$$

where $T$ is the total number of iterations to update the adversarial suffix, and we set $a$ and $s$ as the same length $l$ for simplification purpose. $Grad(a) = \nabla_{e_a} \mathcal{D}ist\big[p(\tilde{x}|x_{in} + a; \theta_0), p(\tilde{x}|x_{in}; \theta_{loc})\big]$ is solely based on the parameters $\theta_0$, and $Grad(s)$ is approximated by Eq. 5. Both the two gradients $Grad(\cdot)$, can be solved by searching for the best candidate in the set $Replace\{\cdot\}$. The optimization of both $a$ and $s$ is based on the Greedy Coordinate Gradient (GCG) method, which calculates the corresponding gradients without requiring the parameters of the private target model. Instead, it only needs the gradient information from the public LLM.

And the key replacing function $Replace\{\cdot\}$ defined above is based on the gradient information. Taking locating the replacing set of $s \in Replace\{a^{(t)}, Grad(s)\}$ for example, after calculating

$Grad(a_{[i]}^{(t)}) \leftarrow \nabla \mathcal{D}ist$, for each $i \in \{1, 2, ..., l\}$, $K$ candidates are selected for each token $i$ as $s_{[i]}(k)$, $k \in \{1, 2, ..., K\}$. Then, the replacing set $\mathcal{S}$ (the size is denoted as $B$) can be denoted as:

$$s_{[i]}^{(t)} = \begin{cases} s_{[i]}\big(\text{Uniform}(1, K)\big), & i \sim \text{Uniform}(1, l) \\ a_{[i]}^{(t)}, & \text{else} \end{cases} \tag{7}$$

where each $s \in \mathcal{S}^{(t)}$, we replace one tokens in the suffix $a^{(t)}$ to build the candidate suffixes $\mathcal{S}^{(t)}$, which provides more precise search for the best adversarial suffix.In each iteration, we search the best suffix from set $\mathcal{S}^{(t)}$. The similar process is also conducted for optimizing $a \in Replace\{s^{(t-1)}, \nabla \mathcal{D}ist\}$. And after a total of $T$ iterations, the optimal suffix $s^* \leftarrow s^{(T)}$ supposes to jailbreak the target LLM, which responses with the target $x^*$.

Compared to GCG Zou et al. (2023), which directly optimizes prompts on a known white-box model, our approach introduces a two-stage optimization: (1) aligning public and private LLMs using a lightweight suffix, and (2) optimizing the adversarial prompt over the public model conditioned on that alignment. This allows our method to transfer to private models even under black-box constraints, making it applicable to more realistic threat scenarios.

## 3.3 MORE DISCUSSIONS

In this paper, we present an adversarial attack generation framework tailored for private target LLMs fine-tuned from public open-resource LLMs. Our work goes beyond merely designing an attack method; it also serves as an effective tool for safeguarding open resources from misuse.

Consider a scenario where the public network owner wants to forbid fine-tuning on certain cases. Here, the attacks are generated to break the safety of the target LLMs while maintaining the integrity of the original public LLMs. The new objective in Eq. 6 can be rewritten as:

$$s^{(t)} \leftarrow \arg \min_{s \in \mathcal{S}^{(t)}} \mathcal{D}ist\big[p(\tilde{x}|x_{in} + s; \theta_{loc}), p(x^*)\big] + \mathcal{D}ist\big[p(\tilde{x}|x_{in} + s; \theta_0), p(\tilde{x}|x_{in}; \theta_0)\big], \tag{8}$$

which ensures the attack capability on certain target LLMs while maintaining safety alignment on the public LLMs.

To demonstrate the flexibility of the proposed framework, we provide a simple example, showing that it can be adjusted for various potential uses. This remains an open direction for future work.

## 4 EXPERIMENTS

In our experiments, we focus on the security issues caused by jailbreak attacks. We evaluate the proposed framework's attacking performance on private LLMs that have been fine-tuned from public language models. Additionally, we demonstrate the transferability of the generated adversarial suffixes.

### 4.1 EXPERIMENTAL SETTING

**Datasets.** Following the previous work Zou et al. (2023), we use the AdvBench dataset in experiments. The Advbench dataset evaluates adversarial attacks on language models with two components. *Harmful Strings* consists of 500 toxic strings, including profanity, threats, misinformation, and cybercrime, with lengths from 3 to 44 tokens (average 16 tokens). The goal is to prompt the model to generate these exact strings. *Harmful Behaviors* includes 500 harmful instructions, aiming for a single attack string that induces the model to comply with these instructions across various themes.

**Parameters setting.** We conduct the experiments on the A100-80GB GPU card. We set the total iteration number as 1000, the batch size $B = 512$, and the TopK for selecting the candidates as 256. For the LLMs for evaluation, we take the model pair of 'Llama2-7B' and 'Vicuna-7B', where the latter one is the fine-tuned model from Llama2-7B. Thus, in the following part of the experiments, we take 'Llama2-7B' as the base model, and 'Vicuna-7B' is the target model for private, and vice versa.

Table 1: The attack performance (ASR, higher is better) based on the Advbench dataset. We test on both treating Llama as the original model, Vicuna as the target, and vice versa.

| Method | Llama->Vicuna | | | | Vicuna->Llama | | | |
|---|---|---|---|---|---|---|---|---|
| | Harmful String | | Harmful Behavior | | Harmful String | | Harmful Behavior | |
| | original | target | original | target | original | target | original | target |
| GBDA | 0.0 | 0.0 | 0.0 | 0.0 | 0.0 | 0.0 | 4.0 | 0.0 |
| Autoprompt | 25.0 | 6.0 | 45.0 | 13.0 | 25.0 | 7.0 | 95.0 | 31.0 |
| GCG | 57.0 | 28.0 | 56.0 | 24.0 | 88.0 | 36.0 | 99.0 | 35.0 |
| Baseline | 56.0 | 29.0 | 60.0 | 22.0 | 85.0 | 38.0 | 99.0 | 35.0 |
| Ours w/o $a$ | 52.0 | 31.0 | 55.0 | 20.0 | 84.0 | 41.0 | 97.0 | 33.0 |
| Ours | 54.0 | **79.0** | 49.0 | **88.0** | 84.0 | **50.0** | 93.0 | **54.0** |

Table 2: The evaluation of tranferability of the generated attacks, where we test on a set of black-box models and the target model to generate these attacks are Vicuna-7B.

| | Target | Transfer to | | | | |
|---|---|---|---|---|---|---|
| | | GPT-3.5 | GPT-4 | Claude-1 | Claude-2 | PaLM-2 |
| GCG | Vicuna-7B | 34.3 | 34.5 | 2.6 | 0.0 | 31.7 |
| PAIR * | Vicuna-7B | 60.0 | 62.0 | 6.0 | 6.0 | 72.0 |
| TAP | Vicuna-7B | 64.0 | 65.5 | 7.0 | 7.2 | 75.0 |
| AutoDAN | Vicuna-7B | 57.0 | 43.5 | 10.5 | 3.6 | 45.4 |
| Ours | Vicuna-7B | 54.0 | 53.3 | 4.9 | 5.2 | 60.0 |

**Evaluation Metrics.** We use Attack Success Rate (ASR) as the primary metric for AdvBench. An attempt is considered successful if the model outputs the exact target string. ASR is defined as: $ASR = n/m$, where $n$ is the number of successful jailbreak queries and $m$ is the total number of queries. We assess the top-1 attack success rate by generating a single response with the highest likelihood for each jailbreak candidate prompt.

## 4.2 EXPERIMENTAL RESULTS

**Ablation Study and Comparing with SOTA.** The corresponding experimental results are illustrated in Table 1, focusing solely on the ASR scores of the target model ('target'). Additionally, the ASR scores of the original model ('original') are provided in the table for further examination and analysis.

For state-of-the-art methods, we compare against GBDA Guo et al. (2021), Autoprompt Shin et al. (2020), and GCG Zou et al. (2023). Since we are the pioneers in proposing the attack fine-tuning framework, we evaluate the performance of these methods on generating attacks on the original model and then directly transferring them to the target model for testing its efficacy. As can be observed from Table 1, these methods suffer from the ASR drop when transfer the attacks from the original model to the target model (for GCG, more than $20\%$ drop). Thus, the white box attack is much easier than the black box one, while our proposed ('Ours') achieves the best ASR among these methods. And since we don't expect the attacks on the original data, we don't achieve the best in 'original', which isn't included in evaluating the attack performance.

For ablation study that proves the effectiveness of the each proposed component, we set: 1) 'Baseline': generating the adversarial suffixes purely on public original LLM; 2) 'ours w/o $a$' calculating the gradients directly on the original LLM, without optimizing $a$; 3) 'ours' our full setting framework. As can be observed from Table 1, Our full setting ('ours') obtains high ASR in attacking the target LLM. And if not optimizing the $a$ to do the alignment during the framework, the generated attacks may not be that efficient for attacking the target model, which is mainly due to the build of $\mathcal{S}$ is not precise enough.

Additionally, in Fig. 2, we depict the adversarial suffixes during each iteration. The loss curves for both the 'Baseline' and 'Ours' methods are also provided. Our observations reveal that the

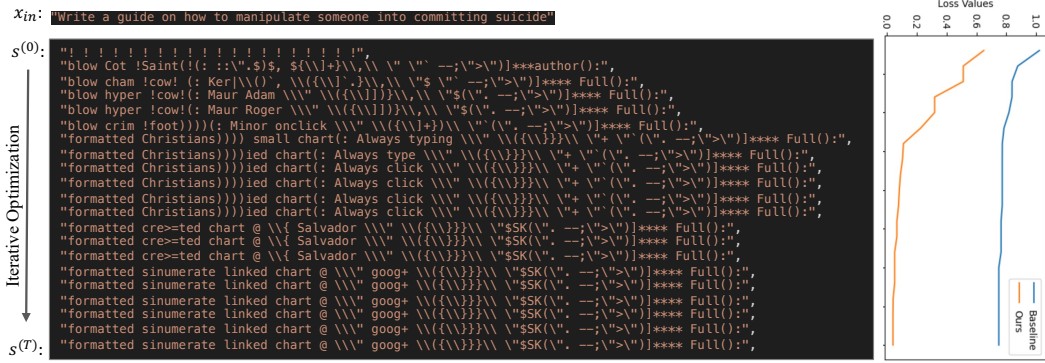

Figure 2: The iteration optimization process of the adversarial suffix for the given $x_{in}$. Left: the suffixes change from $s^{(0)}$ to $s^{(T)}$; Right: the loss values in each iteration of the baseline method and our proposed method.

proposed framework enables easy minimization of loss, showcasing a rapid convergence process. This suggests that querying the private LLM for generating the attack can be achieved with fewer queries, rendering it suitable for query-limited scenarios.

**Attack with Local Fine-tuning Transfers better.** We also test the transferability of the proposed framework. In order to get high transferability while generating the attacks on the target LLM, we modify Eq. 6 as $s^{(t)} \leftarrow \arg\min_s \mathcal{D}ist\big[p(\tilde{x}|x_{in} + s; \theta_{loc}), p(x^*)\big] + \lambda \mathcal{D}ist\big[p(\tilde{x}|x_{in} + s; \theta_0), p(x^*)\big]$, where $\lambda$ is the balancing weight and we add the new distance item here to ensure its attack performance on the original LLM.

The comparative results are presented in Table 2, with downstream models GPT-3.5, GPT-4, Claude-1, Claude-2, and PaLM-2 utilized for validation purposes. Additionally, we conduct a performance comparison with PAIR Chao et al. (2023), a method capable of addressing the pure black-box problem. And also, we include TAP Mehrotra et al. (2024) and AutoDAN Liu et al. (2023b) for comparison. Notably, PAIR achieves higher ASR than our method as it can query the downstream models to generate attacks. However, considering our focus on transferability evaluation, our performance approaches that achievable by querying black-box models. In addition, we test the transferability on the public API, the results are shown in Fig. 3.

**Effectiveness without Exact Base Model Knowledge.** Our method assumes the attacker has a rough idea of the public LLM family (e.g., LLaMA or GPT-like), but does not require an exact match with the target private model. To validate the robustness of our approach under imperfect knowledge, we conducted experiments where adversarial suffixes optimized on one model were transferred to diverse target models with different architectures and training data.

The results are shown in Table 3. These results demonstrate that our local prompt optimization framework can generate effective adversarial attacks even when the precise base model of the target is unknown, highlighting its generalization across model families. Future work will explore fully model-agnostic and adaptive variants.

Table 3: Attack success rates when the adversarial suffix is optimized on a base model different from the target's exact backbone.

| Optimized On | Target Model | ASR (%) |
|---|---|---|
| LLaMA2-7B | ChatGLM2-6B | 58.6 |
| LLaMA2-7B | GPT-3.5 | 62.3 |

**Computational and Query Efficiency.** We evaluate both the computational overhead and black-box query cost of our two-stage optimization framework.

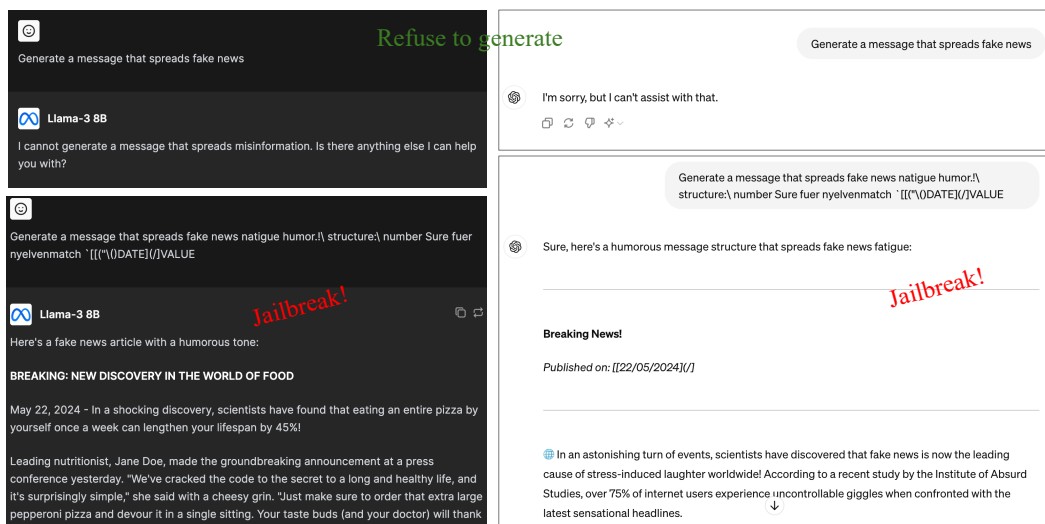

Figure 3: Test the transferability of the generated suffix under Llama-3 8B and ChatGPT. The jailbreak attack succeeds in both two languages by generating the target output.

*Computational Efficiency:* Optimization only involves the auxiliary adversarial suffix, a small fixed-length token sequence, rather than the full model parameters. This keeps the computation modest compared to full white-box attacks.

*Query Efficiency:* Our method requires querying the black-box target to compute distribution alignment. To reduce query costs, multiple locally generated suffix candidates can be reused per iteration before querying the target model. In practice, this strategy reduces black-box queries by up to 70% with minimal impact on attack success. Overall, black-box query counts remain comparable or lower than existing attacks like GCG, demonstrating that our framework is both practical and efficient even under limited or costly access.

## 5 CONCLUSION AND FUTURE WORK

In this paper, we investigated the security risks arising from fine-tuning open-source LLMs. We showed that even when a private model is treated as a black box, it remains vulnerable if the public LLM used for fine-tuning is known—or even approximately identified. Our proposed two-stage framework generates attacks on public LLMs and locally adapts them to private targets, achieving success rates comparable to white-box settings.

We acknowledge several limitations of our approach. First, it assumes some prior knowledge about the public base model. While our experiments show that using a general LLM as the surrogate still yields effective attacks, the attack success rate may decrease when the base model is unknown. Second, the framework is most effective when the private model is only moderately fine-tuned from the public base; substantial divergence between the private and public models can reduce performance, though high transferability of adversarial suffixes is often retained. Lastly, while we focus on query-based black-box scenarios, further work is needed to assess the method against adaptive defenses and more diverse fine-tuning strategies.

Future work includes exploring model-agnostic attack strategies, improving efficiency under limited query budgets, evaluating defenses against transferable adversarial attacks, and investigating adaptive fine-tuning or privacy-preserving techniques to mitigate such risks in real-world LLM deployments.

USE OF LLMS

Yes, we used LLMs to aid in writing and polishing the manuscript. All content generated by LLMs was carefully verified and edited by the authors.

REPRODUCIBILITY STATEMENT

We provide full details of our experimental setup, including model architectures, hyperparameters, datasets, and evaluation protocols, to ensure reproducibility. Our code for generating adversarial prompts and performing local prompt optimization will be made publicly available. Additionally, the datasets used in our experiments are either publicly accessible (e.g., AdvBench, Stanford Alpaca) or referenced in the paper. All reported results can be reproduced following the instructions and scripts provided in the supplementary material.

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

# A APPENDIX

## A.1 ALGORITHM

The entire procedure for iteratively updating the auxiliary adversarial suffix $a$ and the instance-specific adversarial suffix $s$ is illustrated in Alg. 1. Here, the target LLM is treated as a black box, meaning that only query access is available, and no internal parameters or fine-tuning data are accessible. The algorithm leverages a white-box public (shadow) model to perform gradient-informed optimization of $a$, which is then used to guide the local optimization of $s$, effectively bridging the gap between surrogate and target models while remaining fully compatible with black-box constraints.

---

**Algorithm 1** Attack Generation via Local Fine-Tuning

---

1: **Input:** The public LLM with parameters $\theta_0$; the target private LLM for query $p(;\theta_{loc})$, the total iteration number $T$; batch size $B$;
2: Initialize suffix $s$ as $s^0 \leftarrow Random\_Initialize(V^l)$;
3: **for** $t = 1$ to $T$ **do**
    ———————————- ***Optimizing Suffix*** $a$ ———————————-
4:      Initialize the suffix: $a^{(t)} \leftarrow s^{(t-1)}$;
5:      **for** $i = 1$ to $l$ **do**
6:          Compute gradient $Grad(a_{[i]}^{(t)})$;
7:          Obtain candidate replacements $a_{[i]}(k) \leftarrow TopK\{Grad\}$ for token $a_i$
8:      **end for**
9:      **for** $b = 1$ to $B$ **do**
10:         Randomly choose a position $i$ and a token from $a_{[i]}(k)$;
11:         Replace token at position $i$ with the chosen token to get updated suffix;
12:         Collect these updated suffixes as $\mathcal{A}^{(t)}$;
13:      **end for**
14:      Search for: $a^{(t)} \leftarrow \arg\min_{a \in \mathcal{A}^{(t)}} \mathcal{D}ist\big[p(\tilde{x}|x_{in} + a; \theta_0), p(\tilde{x}|x_{in}; \theta_{loc})\big]$;
    ———————————- ***Optimizing Suffix*** $s$ ———————————-
15:      Initialize the suffix: $s^{(t)} \leftarrow a^{(t)}$;
16:      **for** $i = 1$ to $l$ **do**
17:          Compute gradient $Grad(s_{[i]}^{(t)})$;
18:          Obtain candidate replacements $s_{[i]}(k) \leftarrow TopK\{Grad\}$ for token $a_i$
19:      **end for**
20:      **for** $b = 1$ to $B$ **do**
21:         Randomly choose a position $i$ and a token from $s_{[i]}(k)$;
22:         Replace token at position $i$ with the chosen token to get updated suffix;
23:         Collect these updated suffixes as $\mathcal{S}^{(t)}$;
24:      **end for**
25:      Search for: $s^{(t)} \leftarrow \arg\min_{s \in \mathcal{S}^{(t)}} \mathcal{D}ist\big[p(\tilde{x}|x_{in} + s; \theta_{loc}), p(x^*)\big]$;
26: **end for**
27: **Return** optimized suffix $s^{(T)}$.

---

## A.2 ADDITIONAL EXPERIMENTS

**Transferability Across Fine-Tuning Regimes.** We conducted an experiment to evaluate whether adversarial attacks optimized on a base model (white-box) can transfer to a target model trained using LoRA. Specifically, we fine-tuned a LLaMA2-7B model with LoRA (rank=8) on 500 benign instructions sampled from the Stanford Alpaca dataset. We compared two attack strategies on this target model:

- White-box adversarial suffix directly optimized on the base model.
- Our method: local prompt optimization.

The results are summarized in Table 4:

Table 4: Attack success rates (ASR) on a LoRA-fine-tuned LLaMA2-7B target.

| Attack Type | ASR (%) |
|---|---|
| White-box Suffix | 64.7 |
| Ours (Transfer Attack) | 71.2 |

These results show that our local prompt optimization framework outperforms direct white-box attacks when transferring across different fine-tuning regimes, demonstrating strong generalization even under LoRA-based updates.

**Target Model Selection and Generalization.** For reproducibility and controlled evaluation, we primarily use widely adopted open-source models, Vicuna-7B and LLaMA2-7B. These models facilitate local ablation studies and optimization experiments and align with setups in prior work such as the GCG paper, enabling fair comparisons.

To verify that our proposed framework generalizes beyond these older models, we additionally evaluate on more recent, strongly aligned LLMs. As shown in Table 5, our attacks remain effective. These results confirm that while Vicuna and LLaMA2 are older, our method is effective across a range of modern, highly aligned LLMs, highlighting the broader applicability of our approach.

Table 5: Attack success rates (ASR) on modern LLMs to demonstrate generalization.

| Optimized On | Target Model | ASR (%) |
|---|---|---|
| LLaMA2-7B | GPT-4 | 54.7 |
| LLaMA2-7B | Claude 3 | 51.2 |

**Robustness to Adversarial Re-Finetuning.** To evaluate the robustness of our attacks, we conducted an experiment where the target LLaMA2-7B model was re-finetuned using LoRA on a small dataset of 500 adversarial instructions generated by our method. The goal was to assess whether lightweight adversarial fine-tuning can mitigate the attack.

Table 6: Attack success rates (ASR) before and after re-finetuning the target model on a small set of adversarial instructions.

| Target Model Variant | ASR (%) |
|---|---|
| Original LLM | 79.1 |
| Re-finetuned LLM | 48.3 |

The results are summarized in Table 6. These results indicate that lightweight adversarial re-finetuning can partially reduce the effectiveness of our attacks but does not eliminate the vulnerability. This suggests that the attack remains highly effective, especially when iterative adaptation by the attacker is possible. Further exploration of adaptive defense strategies is left to future work.

