# OpenReview forum: "Transferring Jailbreak Attacks from Public to Private LLMs via Local Prompt Optimization"
_ICLR.cc/2026/Conference — Submitted to ICLR 2026_

### Official Review · Reviewer_zurd · 2025-10-25

**Soundness:** 2
**Presentation:** 2
**Contribution:** 2
**Rating:** 4
**Confidence:** 3

**Summary:**

This paper proposes a two-stage local prompt optimization framework for transferring jailbreak attacks from public to private LLMs in black-box settings. The key idea is to introduce an auxiliary adversarial suffix $a$ to align the output distributions between a public base model and a target private model, then optimize the attack suffix $s$ on the aligned public model. The method is evaluated on open-source models (Vicuna, LLaMA2) and proprietary models (GPT-4, Claude), showing high attack success rates comparable to white-box attacks.

The problem is practically relevant given the widespread practice of fine-tuning public LLMs for private deployment. However, the technical contribution is incremental over existing gradient-based methods like GCG, and the theoretical justification for the proposed approximation is insufficient.

**Strengths:**

The paper addresses a realistic threat model where private LLMs are fine-tuned from public models and accessible only via query APIs. This scenario is increasingly common in practice.

The two-stage optimization strategy is intuitive. By first aligning distributions via an auxiliary suffix and then optimizing the adversarial suffix on the aligned model, the method bridges white-box optimization with black-box transfer.

The experimental coverage is broad, including both open-source and proprietary models. The transferability experiments in Table 2 demonstrate that attacks can generalize to various downstream models.

The paper shows that even without knowing the exact base model, using a surrogate model can still achieve reasonable attack success rates (Table 3), which strengthens the practical applicability.

**Weaknesses:**

The core technical contribution is limited. The method essentially combines distribution alignment with GCG-based optimization. The key approximation in Equation 5, $\nabla_{e_s}L_{llm}(s; \theta_{loc}) \approx \nabla_{e_s}L_{llm}(s + a; \theta_0)$, lacks rigorous theoretical justification. The two bullet points provided (PEFT freezes parameters and gradient-based candidate selection has error tolerance) are insufficient to establish when and why this approximation holds. No error bounds or convergence analysis is provided.

The experimental setup has several issues. First, the baseline comparisons primarily use older methods (GBDA, Autoprompt, GCG from 2021-2023). More recent methods like PAIR, TAP, and AutoDAN only appear in the transferability experiments without fair comparison in the main results. Second, the main experiments use Vicuna-7B and LLaMA2-7B, which are relatively old models from 2023. While the appendix includes newer models, the evaluation on state-of-the-art aligned models (e.g., GPT-4o, Claude 3.5 Sonnet) is missing. Third, the computational and query efficiency claims lack detailed analysis. The paper mentions "up to 70% reduction" in queries but provides no breakdown of costs across different stages.

The term "local fine-tuning" is misleading throughout the paper (appearing in Section 3.2 title, Figure 1, Introduction, and Experimental sections). The method does not fine-tune any model parameters but only optimizes prompt suffixes locally. This terminology creates confusion about the actual technical approach. The authors should use more accurate terminology such as "local prompt optimization" consistently throughout the paper.

The evaluation metrics are limited. The paper only reports Attack Success Rate (ASR) based on whether responses contain negative phrases. There is no assessment of attack quality, harmfulness severity, or human evaluation of the generated outputs.

The limitations section acknowledges that the method requires prior knowledge about the base model and is less effective when the private model diverges significantly from the public base. However, the paper does not explore how much divergence can be tolerated or provide guidelines for practitioners.

The defense discussion is superficial. Table 6 in the appendix shows that adversarial re-finetuning reduces ASR from 79.1% to 48.3%, but this is only tested with 500 adversarial examples and lightweight LoRA updates. No other defense mechanisms are explored.

**Questions:**

Can you provide a formal analysis of the approximation error in Equation 5? Under what conditions (e.g., amount of fine-tuning, model architecture) does this approximation hold, and what are the theoretical error bounds?

How does the method perform when the private model undergoes substantial fine-tuning (e.g., full fine-tuning instead of LoRA) or uses a significantly different fine-tuning dataset? Can you provide experimental results on models with varying degrees of divergence from the base?

The query efficiency claim needs more details. How many queries to the target model are required in each iteration? How does this compare to methods like PAIR that directly query the target model? Can you provide a detailed breakdown of query costs?

Why are the newer baseline methods (PAIR, TAP, AutoDAN) not compared in the main experiments (Table 1)? These methods should be included for a fair comparison.

Can you evaluate on more recent and strongly aligned models such as GPT-4o, Claude 3.5 Sonnet, or LLaMA 3.1? The current evaluation on 2023-era models may not reflect the method's effectiveness on state-of-the-art systems.

Have you considered evaluating the quality of generated attacks beyond binary success rates? For example, measuring the actual harmfulness of responses or conducting human studies?

The paper claims this framework can also be used as "an effective tool for safeguarding open resources from misuse" (Equation 8). Can you elaborate on this use case with concrete examples and experiments?

---

### Official Review · Reviewer_eQMX · 2025-10-30

**Soundness:** 2
**Presentation:** 2
**Contribution:** 3
**Rating:** 4
**Confidence:** 3

**Summary:**

The paper proposes a two-stage local prompt optimization attack to transfer jailbreaks from a public LLM to a private fine-tuned LLM accessible. First, an auxiliary suffix is learned to align the public model’s output distribution with the target private model. In second stage, an adversarial suffix is optimized on the public model, using that alignment as a surrogate for target gradients. Experiments report higher ASR on AdvBench and some transfer to closed or target models.

**Strengths:**

* The motivation is clear and grounded in a  public-to-private transfer setting that is relevant for practical security concerns.
* The idea of using an auxiliary alignment suffix to approximate gradients via a public model is promissing.
* The method achieves notable improvements over naive transfer baselines and shows competitive performance against some black-box attacks in certain cases, though gains are not consistent across all settings.

**Weaknesses:**

1. The claims about generality across fine-tuning settings are overstated, as the evidence is limited to a single LoRA-based example in the appendix. Although the introduction mentions broader fine-tuning scenarios (e.g., domain-specific tuning), these are not experimentally validated.
2. The main experiments are limited to a single model pair (LLaMA–Vicuna). This is insufficient to support the broader claims about transferability. Additional pairs such as GPT-NeoX → Dolly, Mistral → Mistral-Instruct, or base → instruct/code models would make the conclusions more robust.
3. The ASR computation method (“judge”) is under-specified. The list of refusal phrases and generation length, are not reported, though these factors can strongly affect results as different models use different refusal phrases or can decide not to answer in later stages [1,2]. Given evidence that rule-based judges are unreliable, an LLM-based judge (e.g., GPT-4 or LLaMa-Guard) would make the evaluation more credible.
4. The approach might be adaptable to other optimization-based attack formulations [3,4], but this potential is unexplored. A brief experiment or discussion would strengthen the paper’s generality claim.
5. The cost comparison is not well controlled. It is unclear whether baselines in Table 1 were reimplemented under the same conditions or taken from prior work. Query budgets and per-example query counts are not reported, making the “comparable efficiency” claim hard to evaluate.
6. The efficiency claims in “Computational and Query Efficiency” part are unclear. Algorithm 1 appears to roughly double the query cost compared to GCG, yet the paper suggests similar or lower cost. Additionally, more detail is needed on how the suffix candidate sets are generated, filtered, and whether this was actually done in experiments.
7. Table 3 lacks baseline comparisons, making it difficult to interpret the reported transfer results in context.

***Minor remarks:***
1. Clarify the meaning of “baseline,” “ours w/o a,” and “ours” in Table 1 in more details, this distinction is currently unclear and hard to follow.
2. The new model results in the appendix are central to the main argument and should be moved into the main text.
3. Define the term “black-box” consistentlyas in the literature it can sometimes mean query-only access, other times query + logits.
4. The asterisk (*) in Table 2 is not explained anywhere.
5. The results on the right half of Table 1 are somewhat inconsistent with the paper’s narrative, since Vicuna is a fine-tuned derivative of LLaMA rather than the other way around. The interpretation of transfer direction and its implications for fine-tuning robustness should therefore be clarified.
6. The discussion of Table 2 notes that the method underperforms PAIR but omits that it also underperforms TAP, this should be acknowledged for completeness.
7. Section 3.3 reads like the beginning of a separate chapter rather than a continuation of the current discussion. You might consider reframing its opening or adding a short transition to better connect it with the previous sections.

[1] Zhou, Y., & Wang, W. (2024). Don’t Say No: Jailbreaking LLM by Suppressing Refusal

[2] Ran, D., Liu, J., Gong, Y., Zheng, J., He, X., & Cong, T. et al. (2025). JailbreakEval: An Integrated Toolkit for Evaluating Jailbreak Attempts Against Large Language Models

[3] Jia, Xiaojun; Pang, Tianyu; Du, Chao; Huang, Yihao; Gu, Jindong; Liu, Yang; Cao, Xiaochun; Lin, Min (2024). Improved Techniques for Optimization-Based Jailbreaking on Large Language Models

[4] Andriushchenko, M., Croce, F., Flammarion, N. (2025). Jailbreaking Leading Safety-Aligned LLMs with Simple Adaptive Attacks

**Questions:**

1. Have you tested transferability between different versions within the same model family (e.g., LLaMA 2 → LLaMA 3)? Although it is not technically finetuning, these models share many components and it would be interesting to see their transfer.
2. What happens when transferring between different model sizes (e.g., 3B → 7B)?
3. How do you interpret these results in light of findings that fine-tuning often reduces robustness [6]?
4. Could you provide the attack results without the auxiliary weight (Table 2) to isolate its contribution?
5. Have you considered using ensemble GCG (as described in section 3.2 of [7]) in this setting? It may improve transfer, especially when the target family is unknown.
6. Have you tested the attacks against any defense mechanisms? Are they more or less likely to be detected than baseline attacks?
7. Did you keep the models’ default system prompts (e.g., “You are a helpful AI assistant”) during all experiments, or were they overridden? This choice can meaningfully affect jailbreaking success rates.

[5] Qi, X., Zeng, Y., Xie, T., Chen, P.-Y., Jia, R., Mittal, P., & Henderson, P. (2023). Fine-Tuning Aligned Language Models Compromises Safety, Even When Users Do Not Intend To!

[6] Zou, A., Wang, Z., Carlini, N., Nasr, M., Kolter, J. Z., & Fredrikson, M. (2023). Universal and Transferable Adversarial Attacks on Aligned Language Models.

---

### Official Review · Reviewer_P6Bv · 2025-10-31

**Soundness:** 1
**Presentation:** 1
**Contribution:** 1
**Rating:** 2
**Confidence:** 5

**Summary:**

This paper aims to jailbreak black-box LLMs that were originally finetuned from some open-source LLMs. The authors design a transfer attack that synthesizes jailbreak prompts based on the original open-source LLMs to attack the black-box finetuned ones. The proposed attack was compared with some very old jailbreak baselines, but unfortunately could not beat the black-box PAIR attack in both jailbreak effectiveness and efficiency.

**Strengths:**

**None.**

This paper contains two major flaws that make it impossible for me to vote for acceptance:
- The proposed attack implicitly requires access to the full next-token sampling distribution of the targeted LLM, which, however, is basically impossible in real-world LLM services.

- Both the query efficiency and the attack effectiveness of the proposed black-box attack are significantly weaker than those of other existing black-box attacks with fewer assumptions.

Please see **Weaknesses** for details.

**Weaknesses:**

1. **(Major flaw 1)** In Line 213, the authors state that "they can implement the distance function $Dist(\cdot)$ as the cross entropy loss". Since the authors do not discuss this distance function further, I have to assume that they directly implement $Dist(\cdot)$ as the cross-entropy loss in all experiments in their paper. If this is the case, it means the authors implicitly assume that the attacker can access **the full next-token sampling distribution of the targeted LLM**, since the cross-entropy loss needs to be calculated on the output logits/distributions of the targeted model. However, due to privacy issues [r1], real-world LLM black-box services typically do not provide access to the full next-token sampling distributions to end users. For example, OpenAI only allows access to the log probabilities for up to the $20$ most likely tokens [r2]. This issue significantly shrinks the practicality of the proposed attack.

2. **(Major flaw 2)** The proposed black-box attack is significantly weaker than the PAIR attack, which is a black-box attack that this paper uses as a baseline, in both query efficiency and attack effectiveness. Furthermore, the PAIR attack does not even need to assume that the targeted private model is fine-tuned from an open-source model. As a result, I cannot see any advantages of the proposed attack. Specifically, for query efficiency, the proposed attack needs to query the targeted model $O(1000)$ times (see Line 320; I am not sure if it is $1000$ or $2000$), while the PAIR attack only needs to query the targeted LLM up to $20$ times to find an effective jailbreak prompt (see the original paper [r3] of the PAIR attack). For attack effectiveness, Table 2 clearly shows that the proposed attack is significantly weaker than the PAIR attack.

3. The attack model the authors consider is very narrow and not practical at all. I do not think it is very reasonable or practical to assume that an arbitrary black-box targeted LLM is fine-tuned from an open-source LLM. The authors need to provide more real-world evidence to justify this.

4. In Eq. (3), the authors claim that "$p(x^*_{[n+1:n+H]})$ is the target output logit values". Is this the logit of the original model or the fine-tuned one? Besides, it seems that this is not a logit conditional on an input prompt, is it? So why would you want to align a conditional logits/distribution to this unconditional one under the $Dist(\cdot)$ function? From my perspective, such an operation is meaningless.

5. The GBDA baseline considered is too old and weak. [r4] has already shown that the GCG attack is much stronger than the GBDA attack. The authors should consider more advanced jailbreak attack baselines such as [r5, r6, r7].

6. The authors aim to attack models fine-tuned from open-source models. However, their experiments relevant to open-source LLMs are conducted on only two models: Vicuna-7B and Llama-2-7B. Both of these open-source models are too old and too weakly aligned, so I do not think the empirical conclusions drawn from these models can be generalized.

**References**

[r1] Carlini et al. Stealing Part of a Production Language Model. ICML 2024.

[r2] https://platform.openai.com/docs/api-reference/responses/create#responses_create-top_logprobs

[r3] Chao et al. Jailbreaking Black Box Large Language Models in Twenty Queries. arXiv 2023.

[r4] Zou et al. Universal and Transferable Adversarial Attacks on Aligned Language Models. arXiv 2023.

[r5] Sadasivan et al. Fast Adversarial Attacks on Language Models In One GPU Minute. ICML 2024.

[r6] Hayase et al. Query-Based Adversarial Prompt Generation. NeurIPS 2024.

[r7] Andriushchenko et al. Jailbreaking Leading Safety-Aligned LLMs with Simple Adaptive Attacks. ICLR 2025.

**Questions:**

See **Weaknesses**.

---

### Official Review · Reviewer_8BPu · 2025-11-01

**Soundness:** 3
**Presentation:** 1
**Contribution:** 2
**Rating:** 4
**Confidence:** 3

**Summary:**

In this paper, the authors propose a two-stage optimization-based attack to threaten the black-box private LLMs. Experiments have demonstrated their transferability and effectiveness between open-source and closed-source models.

**Strengths:**

1 This paper is easy to follow.

2 The attack is effective for both closed-source and open-source models.

3  The soundness of the proposed method is good.

**Weaknesses:**

1 The author appears to have confused the "\citep" command with the "\cite" command. Based on my observation, it seems the incorrect command has been used throughout the entire paper.

2 Some typos exist in this paper.  For example, in Line 280, there should be a space before 'In each iteration'.

3 No normalization to ensure the readability of the crafted suffix. This makes the attack super fragile to the PPL-based detection method [1].

4 The baselines for comparison are old. An example is that GBDA, Autoprompt, and GCG are attacks that were proposed before the year of 2023. I suggest authors compare their attack with more SOTA attacks, such as AutoDAN-turbo [2].

5 In Table 2, the authors compared their proposed attacks with the query-based and optimization-based attacks. However, as far as I know, context-based attrack like ICA [2] and I-FSJ [3] is another kind of black-box attack, I suggust authors add them into Table 2 as baseline methods.

6 The robustness of the attack against defense is not evaluated, like smoothLLM [5], PAT [6] and RPO [7].

[1] Baseline Defenses for Adversarial Attacks Against Aligned Languge Models

[2] Autodan-turbo: A lifelong agent for strategy self-exploration to jailbreak llms

[3] Jailbreak and guard aligned language models with only few in-context demonstrations

[4] Improved few-shot jailbreaking can circumvent aligned language models and their defenses

[5] SmoothLLM: Defending Large Language Models Against Jailbreaking Attacks

[6] Fight Back Against Jailbreaking via Prompt Adversarial Tuning

[7] Robust prompt optimization for defending language models against jailbreaking attacks

**Questions:**

1 I think Section 3.3 is confusing. Why do the public owners want to forbid fine-tuning in certain cases?

2 In Equation 8, the $s^{(t)}$ minmizes the formulation $Dist[p(\tilde{x}|x_{in}+s;\theta_0),p(\tilde{x}|x_{in};\theta_0)]$. Does this formulation indicate that $s$ has no impact on the output of the target LLMs?

---

### Meta-Review · Area_Chair_X4kh · 2025-12-04

**Summary:**

The paper proposes a two-stage local prompt optimization framework to transfer jailbreak attacks from public to privately fine-tuned LLMs, addressing a black-box setting where only API access is available. The method introduces an auxiliary adversarial suffix to align output distributions between public and private models, enabling gradient-based optimization without accessing private parameters. Reviewers acknowledged the practical relevance of the problem and the clarity of motivation. However, the paper received mixed evaluations, with significant concerns raised about technical soundness, experimental rigor, and contribution novelty.

**Reviewer Scores:**

N/A

---

### Decision · Program_Chairs · 2026-01-26

Reject